# Feline Erythrocytic Osmotic Fragility in Normal and Anemic Cats—A Preliminary Study

**DOI:** 10.3390/vetsci12030236

**Published:** 2025-03-03

**Authors:** Purin Lophaisankit, Kunanon Boonyok, Jaruwan Khonmee, Chatchanok Udomtanakunchai, Chollada Sodarat, Kannika Phongroop, Worapat Prachasilchai

**Affiliations:** 1Faculty of Veterinary Medicine, Chiang Mai University, Chiang Mai 50100, Thailand; purin_lo@cmu.ac.th (P.L.); jaruwan.khonmee@cmu.ac.th (J.K.); chollada.s@cmu.ac.th (C.S.); kannika.p@cmu.ac.th (K.P.); 2Faculty of Veterinary Medicine, Kasetsart University, Bangkok 10900, Thailand; kunanonboonyok@gmail.com; 3Department of Radiologic Technology, Faculty of Associated Medical Sciences, Chiang Mai University, Chiang Mai 50200, Thailand; chatchanok.u@cmu.ac.th; 4Research Center in Bioresources for Agriculture, Industry and Medicine, Chiang Mai University, Chiang Mai 50100, Thailand

**Keywords:** erythrocyte osmotic fragility, anemia, red blood cell, forward scatter characteristics, side scatter characteristics

## Abstract

This study assessed red blood cell membrane strength and characteristics in non-anemic and anemic cats using the osmotic fragility test and flow cytometry. Blood samples from 18 non-anemic and 18 anemic cats, divided into adults and seniors, were analyzed for complete blood counts, blood chemistry, the osmotic fragility test, and flow cytometry. The findings showed no significant differences in osmotic fragility between non-anemic and anemic cats in either age group. In flow cytometry analysis, normal senior cats had higher forward scatter than senior anemic cats, while normal adult cats had higher side scatter than anemic adult cats. These results suggest that, while osmotic fragility remains consistent, red blood cell size and density differences vary with age and health status, highlighting the need for further research on red blood cell characteristics in different diseases.

## 1. Introduction

Red blood cells (RBCs) are unique structures in vertebrates that, upon maturation, lose the ability to perform ribosomal protein synthesis or mitochondrial phosphorylation. Their fatty acid and phospholipid compositions vary across species [1]. Evaluating the RBC membrane is critical for understanding its functional integrity, particularly in pathological conditions. Osmotic fragility (OF) is the resistance of RBC hemolysis to osmotic changes used to assess RBC friability, the susceptibility of red blood cells to break or rupture under mechanical stress or osmotic pressure, which may be abnormal in various diseases [2,3,4]. Several methods exist for measuring OF, including the classic OF test, the eosin-5′-maleimide dye-binding test, the flow cytometric OF test, and the cryohemolysis test [5]. The classic OF test, a widely used method, involves incubating whole blood in a hypotonic sodium chloride (NaCl) solution at room temperature [6]. Exposure to the hypotonic solution causes water to diffuse into RBCs, leading to swelling, leakage, and eventual lysis. Hemoglobin (Hgb) released into the supernatant is measured spectrophotometrically, with the results expressed as the percentage of completely lysed blood in distilled water. The normal OF curve is sigmoid, and the mean osmotic fragility (MOF)—the NaCl concentration at which 50% of RBCs are lysed—typically occurs around 0.5% NaCl in healthy animals. OF is generally inversely correlated with the normal size of RBCs across species [7] and can be used to evaluate the rate of hemolysis [8].

Poikilocytosis, the presence of abnormally shaped RBCs in the blood, provides significant diagnostic information [9]. Alterations in RBC morphology can aid in identifying underlying diseases in both humans and animals [10,11,12]. In addition to pathological causes, erythrocyte morphology can change due to physiological factors, such as aging [13]. Flow cytometry offers another valuable tool for analyzing RBC populations by utilizing lasers to produce scattered light signals. These signals are detected and analyzed, with forward scatter (FSC) indicating relative cell size and side scatter (SSC) reflecting internal complexity or granularity [14].

Feline anemia is a common condition that can develop among many other conditions, such as hemorrhage, hemolysis, bone marrow suppression, chronic inflammation, tumors, endocrine diseases, and nephropathies, among others. Previous studies have found that increased erythrocytic OF, leading to hemolysis, has been reported in various conditions, including immune-mediated hemolytic anemia in cats [15], anemia in domestic shorthair and other cat breeds [3], and hereditary RBC membrane defects in Abyssinian and Somali cats [16]. Fragile RBCs have also been associated with conditions such as chronic intermittent hemolysis, macrocytic-hypochromic and variably regenerative anemia, splenomegaly, hyperglobulinemia, lymphocytosis, hyperbilirubinemia, stress, hyperlipidemia, hyperglycemia, microcytosis due to intestinal parasitism, intoxication with beta-acetylphenylhydrazine, and elevated serum hepatic enzyme activities [3,8,17,18,19,20,21,22]. This study aims to evaluate RBC membrane strength using the classic osmotic fragility test in various groups of cats, including healthy and anemic individuals. Additionally, the study seeks to compare RBC size and complexity across these groups, contributing valuable insights into feline hematology and the diagnostic potential of RBC characteristics in identifying underlying conditions.

## 2. Materials and Methods

### 2.1. Sample Population

A total of 36 client-owned domestic shorthair cats of all genders more than 1 year old from the Small Animal Hospital, Chiang Mai University were included in our study between May and December 2023. Animals were divided into 2 groups: normal (non-anemic) cats (*n* = 18) and anemic cats (*n* = 18). Each group was divided into 2 subgroups: adults (1–5 years old, *n* = 9) and seniors (>6 years old, *n* = 9). Therefore, the cats were categorized into four groups: normal adult (NA), normal senior (NS), anemic adult (AA), and anemic senior (AS).

All cats underwent a physical examination, including temperature measurement, observation of general appearance, hydration status, body condition score, mentation, posture, thoracic auscultation, abdominal palpation, and blood collection. If the physical examination results showed no clinical signs and hematologic parameters were within normal limits [23], cats were categorized into the non-anemic group. There was no infectious disease screening (e.g., FeLV, FIV, or other region-specific pathogens) of individuals as part of the initial study. Some further investigations, including diagnostic imaging or more specific tests, were not performed for every individual with abnormalities.

Cats were included in the anemic group if at least one of the following clinical signs were present: pale mucous membranes, loss of appetite, weight loss, depression, tachycardia, hematochezia, and melena, along with laboratory findings showing a hematocrit less than 30%.

This study received ethical approval from the Faculty of Veterinary Medicine, Chiang Mai University Animal Care and Use Committee (FVM—ACUC). The reference number is S11/2566.

### 2.2. Sample Collection

Blood samples (1.5 mL) were collected from non-fasting, well-hydrated cats and then divided into 0.5 mL in an ethylenediaminetetraacetic acid (EDTA) tube and 1 mL in a heparin tube. Blood samples were chilled to 4 °C after collection and shipped to the laboratory for analysis within 24 h after collection.

### 2.3. Laboratory Testing

Routine clinical laboratory techniques were used to perform complete blood counts (Dymind DF56VET, Dymind Biotechnology, Shenzhen, China) and microscopic blood smear (Dip Quick Stain, M&P impex, Bangkok, Thailand) evaluations, including RBC morphology, platelet counts, and blood parasites, performed by the scientists of the Small Animal Hospital laboratory using the EDTA tube. The blood in the heparin tube was analyzed for blood chemistry panels (Sysmex DX-3010, Sysmex corporation, Kobe, Japan), including alkaline phosphatase (ALP), alanine transaminase (ALT), blood urea nitrogen (BUN), creatinine, triglycerides (TG), total cholesterol (Chol), and blood glucose. A blood smear was performed to evaluate blood parasites, such as *Mycoplasma haemofelis*, *Babesia felis*, etc., to evaluate any abnormalities of erythrocyte morphology and to evaluate platelet counts. The platelet count was assessed using a blood smear examined under a 100× oil immersion lens. Platelet numbers were estimated using the following formula: platelet count (per µL) = number of cells per 100× oil field × 20,000. A platelet count within the normal range was considered adequate.

### 2.4. Osmotic Fragility Test

The leftover blood from the EDTA tube was centrifuged at 7000 rpm for 1 min. The classic OF test of the erythrocyte was performed by preparing a series of 1 mL NaCl tubes containing concentrations of 0.9, 0.7, 0.6, 0.5, 0.4, 0.2, and 0.1% NaCl. Packed red cells (10 µL) were diluted with 300 µL of 0.9% NaCl. A 10 µL volume of the diluted solution was added to each tube, followed by incubation at room temperature for 3 min and centrifugation at 7000 rpm for 1 min. The supernatant’s optical density was measured using a spectrophotometer (Shimadzu UV-2700, UV-Vis spectrophotometer, Kyoto, Japan) at 415 nm. For each tube, the hemolysis percentage was calculated. The mean OF was determined from a hemolysis curve [3,16].

### 2.5. Flow Cytometry Test

The flow cytometry test was performed by diluting 5 µL of packed red cells with 1 mL of 0.9%, and then forward scatter characteristics (FSC) and side scatter characteristics (SSC), which refer to size and density, respectively, were measured by flow cytometry (CytoFLEX Flow Cytometer, Beckman Coulter, Brea, CA, USA).

### 2.6. Statistical Analysis

The following data were collected in Excel (Microsoft Corporation, Redmond, Washington). Descriptive statistics were used for analysis. Continuous variables were expressed as mean ± standard deviation (SD). The statistical analysis was performed using OriginPro 2015 software (OriginLab Corporation, Northampton, MA, USA). An independent *t*-test was used to compare continuous variables between groups and within groups. The correlation between blood profiles and laboratory results was determined by linear regression. A *p*-value < 0.05 indicated statistical significance. A correlation coefficient (r) > 0.7 was considered a strong relationship, between 0.4 to 0.7 was considered a moderate relationship, and <0.4 was considered a weak relationship [24].

## 3. Results

### 3.1. Animals

A total of 36 client-owned domestic shorthair cats from the Small Animal Hospital, Chiang Mai University were included in this study between May and December 2023. All cats underwent clinical assessment for dehydration prior to blood collection, and none were found to exhibit signs of dehydration. The NA group (*n* = 9) comprised five males and four females, with an average age of 2.2 ± 1.2 years. The SA group (*n* = 9) included three males and six females, with an average age of 8.1 ± 2.3 years. The AA group (*n* = 9) consisted of four males and five females, with an average age of 2.0 ± 1.4 years. Lastly, the SA group (*n* = 9) comprised three males and six females, with an average age of 9.8 ± 2.8 years. The average ages were expressed as mean ± SD.

Several etiologies contributing to anemia were considered within this study. In the anemic adult group, anemia was predominantly caused by infectious disease (55.55%, 5/9). Conversely, in the anemic senior group, anemia was prominently associated with kidney disease (44.44%, 4/9), as shown in Table 1.

### 3.2. Laboratory Tests

All normal cats exhibited hematologic parameters that were mostly within normal ranges [23], except for the platelet count, which appeared to decrease on the analyzer but was adequate upon blood smear evaluation. On the other hand, anemic cats exhibited a diverse range of hematologic parameters, with six cats (33.33%, 6/18) displaying a decreased platelet count upon platelet examination.

Mean hematologic parameters, including hematocrit (Hct), hemoglobin (Hgb), red blood cell counts (RBCs), mean corpuscular volume (MCV), mean corpuscular hemoglobin (MCH), mean corpuscular hemoglobin concentration (MCHC), white blood cell counts (WBCs), and platelet counts (PLTs) of cats from each group are displayed in Table 2. All groups of cats showed a decreased platelet count when assessed by the analyzer. However, blood smear analysis revealed an adequate platelet quantity in all cats from the normal group (100%, 18/18) and in 12 cats from the anemia group (66.66%, 12/18).

The majority of normal cats (94.44%, 17/18) exhibited blood chemistry parameters within the normal range; however, an exception was observed in the normal senior group, where one cat displayed mildly elevated ALT levels.

Six anemic cats (33.33%, 6/18) displayed elevated levels of BUN and creatinine. Notably, within the anemic senior group, one cat exhibited a severe elevation of ALT.

Mean blood chemistry parameters, including BUN, creatinine, ALT, ALP, total protein, albumin, cholesterol, triglycerides, and glucose, of the cats in each group are displayed in Table 3.

### 3.3. Osmotic Fragility Test

The 50% osmotic fragility value revealed no statistically significant difference between groups, and the mean 50% osmotic fragility is shown in Figure 1.

### 3.4. Flow Cytometry Test

The forward scatter characteristics (FSCs) of each group showed significant differences in the mean FSC between the normal senior group and the normal adult group (*p* = 0.02864), with the normal senior group exhibiting higher FSC values, and the normal senior group also demonstrated higher FSCs than the anemic senior group (*p* = 0.0486), as shown in Figure 2. 

The side scatter characteristic (SSC) of each group showed a significant difference in mean SSC between the normal adult group and the anemic adult group (*p* = 0.048), with the normal adult group exhibiting higher SSC values than the anemic adult group, as shown in Figure 3.

Linear regression analyses demonstrated the relationship between osmotic fragility and the forward scatter characteristic of each group, revealing a statistically significant positive correlation in the anemic senior group (AS), with *p* = 0.0103, r = 0.79, and R^2^ = 0.58, as shown in Figure 4.

## 4. Discussion

A total of 36 client-owned domestic shorthair cats were enrolled in the study and categorized into two groups: non-anemic cats and anemic cats. All the cats in the non-anemic group exhibited no clinical signs and demonstrated nearly all typical hematologic and blood chemistry parameters within normal ranges, except one cat in the senior group that displayed mildly elevated ALT. These findings suggest the possibility of underlying conditions that were not fully explored. Every group of cats showed thrombocytopenia and increased triglycerides. However, upon examination of platelet smears, an adequate quantity of platelets was observed, indicating that the deviation in the platelet count may not be indicative of an actual platelet deficiency, and these results should be interpreted with caution because the fasting statuses of patients at the time of sample collection were not standardized in this study, which may explain the elevated triglyceride levels observed across groups. Within the anemic group, several abnormalities were observed in hematologic and blood chemistry parameters. Notably, three cats that exhibited a decrease in platelet quantity upon examination were associated with *Mycoplasma* spp. infection. These findings are consistent with those reported in previous studies [25,26,27]. However, a major limitation of this study is the lack of confirmation of mycoplasmosis through polymerase chain reaction (PCR). Diagnoses in three cats were based on blood smear evaluation, which has low sensitivity and specificity. As a result, some cats classified as negative for mycoplasmosis may have had low parasitemia. We recommend that future studies incorporate PCR testing to accurately confirm mycoplasmosis status.

In terms of blood chemistry, six cats in the anemic group (33.33%, 6/18) demonstrated azotemia, indicating that a notable number of them were associated with kidney disease. A significant limitation of this study is the absence of infectious disease testing (e.g., FeLV, FIV, or other region-specific pathogens) in the initial study design, which could have affected the inclusion of cats with latent or subclinical infections. 

Our research findings revealed a mean osmotic fragility (OF) of 0.586 ± 0.086 in the normal adult group, while the normal senior group exhibited a higher mean OF of 0.630 ± 0.063. Remarkably, our results indicate a higher mean OF than reported in a previous study [3,16]. Conversely, within the anemic group, the mean OF for anemic adults was 0.656 ± 0.064, and anemic seniors had a mean OF of 0.638 ± 0.088, in close agreement with a prior study [3,16]. Nevertheless, our results on osmotic fragility revealed no statistically significant differences between groups, as shown in Figure 1. This variation may be attributed to the diversity observed in osmotic fragility within the normal group, particularly in the normal senior group, which exhibited higher osmotic fragility than the normal adult cats. Contrarily, in the anemic group, some cats demonstrated very high osmotic fragility, while others exhibited low osmotic fragility. This variation suggests that osmotic fragility differs between diseases. Therefore, we suggest that further studies are necessary, and it is advisable to increase the population size for each anemia-inducing disease.

A thorough review of the literature shows that kidney disease leads to imbalances in calcium phosphate, acid-base chemistry, and electrolyte levels due to uremia, which, in turn, affects red blood cell shape and survival [28,29]. Secondary hyperparathyroidism, associated with CKD, arises from underlying hyperphosphatemia, low calcium, and reduced calcitriol levels. Parathyroid hormone (PTH) affects the osmotic fragility of RBCs by enhancing calcium entry into cells [30]. This increased calcium influx stimulates Ca^2+^-activated ATPase, resulting in ATP depletion and erythrocyte fragmentation. Elevated PTH levels also increase RBC calcium concentrations, leading to the cross-linking of membrane proteins, which disrupts red blood cell structure and stability, ultimately causing cell lysis [31]. Secondary immune-mediated hemolytic anemia has been documented in cats with FeLV infection, hemobartonellosis, and lymphoma [32,33,34], all of which are associated with increased OF. Moreover, FeLV infection, inflammation, and neoplastic disease [35,36,37] may reduce the lifespan of RBCs in cats.

Using flow cytometry analysis, we found that the normal senior group exhibited a higher mean FSC than the normal adult group. This result contrasts with a previously conducted study in humans [38], which reported that the normal adult group had a higher FSC than the normal senior group. In terms of SSC, our findings revealed that the normal adult group exhibited a higher SSC than the anemic adult group. This finding is related to a prior study that reported a correlation between side scatter characteristics and the hemoglobin concentration of red blood cells [39]. In the senior group, we observed insignificant differences, which could be due to the predominance of kidney disease in the anemic senior group, whereas the anemic adult group was primarily associated with infectious disease, which might have a more pronounced effect on the hemoglobin concentration.

Linear regression analyses revealed a positive correlation between FSC and OF, indicating that a larger RBC size is associated with higher osmotic fragility. This finding is similar to those of a previous human study, which reported that larger RBCs are associated with increased fragility, rendering them more susceptible to damage and subsequent lysis [38,40]. In contrast with previous studies [17,18,19], our investigation did not identify a significant relationship between OF and triglycerides, cholesterol, or glucose levels.

The results could contribute to building a database of OF in various diseases for future studies. Additionally, these findings could aid in the development of a rapid OF test, a screening tool for each cause of anemia in patients, and the correlation between OF and RBC size and density may be used to create a predictive model for OF.

## 5. Conclusions

Our study revealed a higher mean OF in normal cats than reported in previous studies. However, both normal and anemic cats exhibited similar osmotic fragilities. We found that a larger RBC size is associated with higher osmotic fragility. Further studies of various anemia-inducing diseases are suggested.

## Figures and Tables

**Figure 1 vetsci-12-00236-f001:**
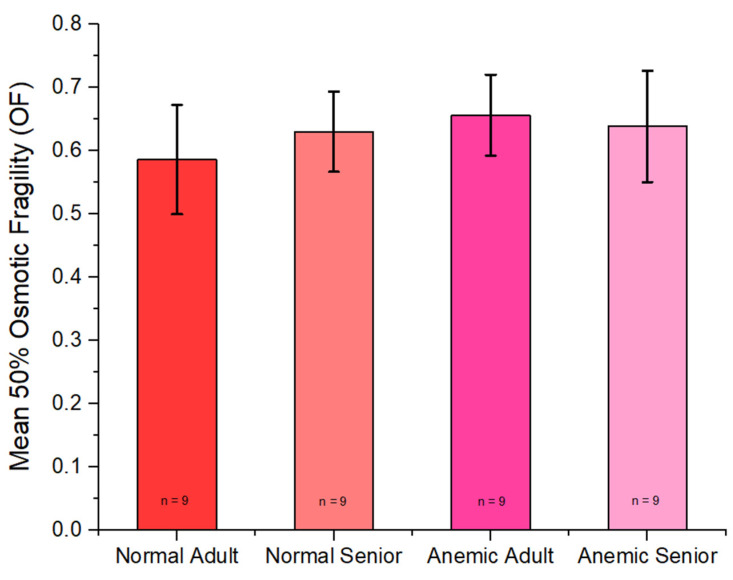
Mean 50% osmotic fragility (OF) of the normal adult, normal senior, anemic adult, and anemic senior groups. Error bar represents SD.

**Figure 2 vetsci-12-00236-f002:**
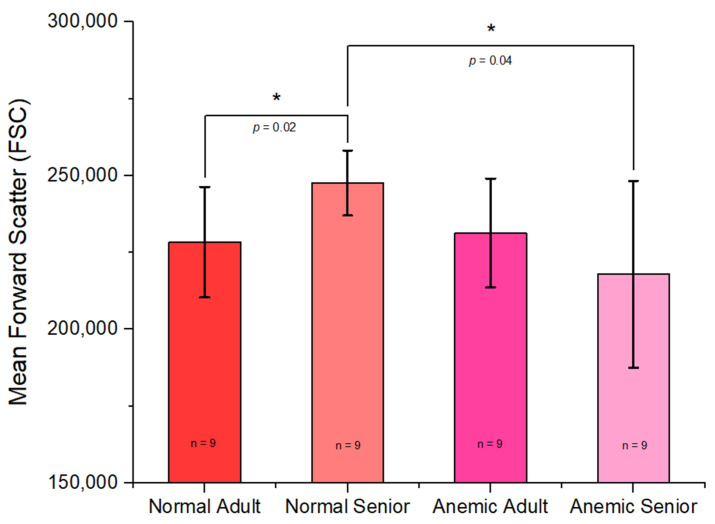
Mean forward scatter characteristics (FSC) of the normal adult, normal senior, anemic adult, and anemic senior groups. Error bar represents SD. * Statistically significant (*p*-value < 0.05).

**Figure 3 vetsci-12-00236-f003:**
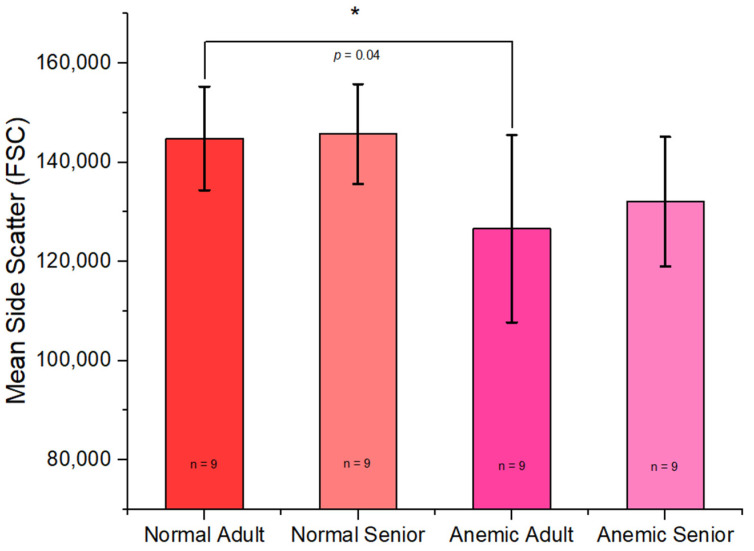
Mean side scatter characteristics (SSC) of the normal adult, normal senior, anemic adult, and anemic senior groups. Error bar represents SD. * Statistically significant (*p*-value < 0.05).

**Figure 4 vetsci-12-00236-f004:**
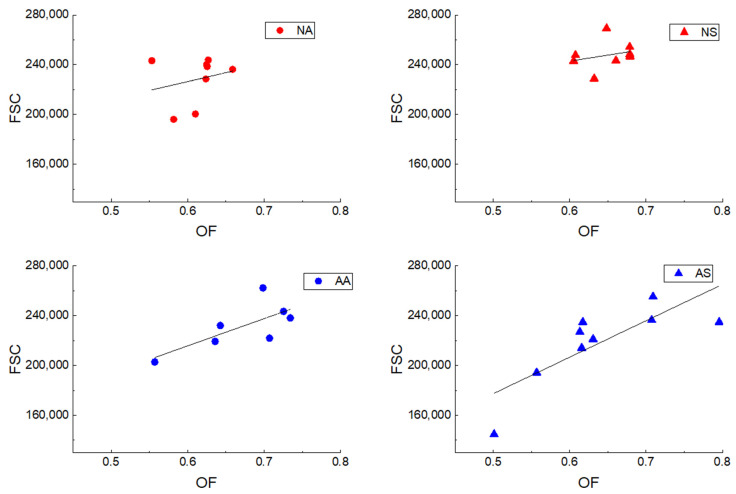
Linear regression analyses demonstrated the relationship of osmotic fragility and forward scatter characteristic of the normal adult group (NA), normal senior group (NS), anemic adult group (AA), and anemic senior group (AS).

**Table 1 vetsci-12-00236-t001:** The cause of anemia in the anemic adult (*n* = 9) and anemic senior (*n* = 9) groups.

Group	Disease	*n*
Anemic adult	Polycystic kidney disease (PKD)	1
	Chronic kidney disease (CKD)	1
	Blood loss (hit by car)	1
	Feline leukemia virus (FeLV)	3
	Blood parasite (*Mycoplasma* spp.)	1
	Feline immunodeficiency virus (FIV)	1
	Lymphoma (CHOP treatment)	1
Anemic senior	Chronic kidney disease (CKD)	4
	Blood loss (hit by car)	1
	Blood loss (post-operative anemia)	1
	Blood parasite (*Mycoplasma* spp.)	2
	Unknown cause	1

**Table 2 vetsci-12-00236-t002:** Mean hematologic parameters of normal adult, normal senior, anemic adult, and anemic senior cats.

Parameter (Unit)	Normal Adult	Normal Senior	Anemic Adult	Anemic Senior	References Interval
Hct (%)	43.33 ± 3.42 ^a^	42.88 ± 1.81 ^a^	19.60 ± 8.76 ^b^	22.00 ± 6.93 ^b^	30–45
Hgb (g/dL)	14.40 ± 1.24 ^a^	14.30 ± 0.90 ^a^	6.08 ± 2.85 ^b^	8.50 ± 5.27 ^b^	9.8–15.4
RBCs (×10^6^/uL)	8.67 ± 0.60 ^a^	8.47 ± 0.89 ^a^	3.89 ± 1.84 ^b^	4.84 ± 2.14 ^b^	5–10
MCV (fL)	50.23 ± 4.38	51.04 ± 3.89	51.07 ± 6.28	51.89 ± 13.38	39–55
MCH (pg)	16.62 ± 0.83	17.01 ± 0.89	15.62 ± 2.11	18.89 ± 9.27	13–17
MCHC (g/dL)	33.21 ± 2.33	33.39 ± 1.23	30.63 ± 2.81	29.33 ± 6.87	30–36
WBCs (×10^3^/uL)	11.43 ± 6.44 ^a^	9.73 ± 2.81 ^a^	24.76 ± 9.55 ^b^	32.69 ± 29.04 ^b^	5.5–19.5
PLTs (×10^3^/uL)	255.77 ± 54.33 ^a^	235.88 ± 51.53 ^a^	143.00 ± 116.83 ^b^	171.44 ± 133.45 ^b^	300–800

Data are expressed as mean ± standard deviation. Different letters mean statistically significant difference (*p*-value < 0.05). Abbreviations: Hct = hematocrit, Hgb = hemoglobin, RBCs = red blood cells, MCV = mean corpuscular volume, MCH = mean corpuscular hemoglobin, MCHC = mean corpuscular hemoglobin concentration, WBCs = white blood cells, and PLTs = platelets.

**Table 3 vetsci-12-00236-t003:** Mean blood chemistry parameters of normal adult, normal senior, anemic adult, and anemic senior cats.

Parameter (Unit)	Normal Adult	Normal Senior	Anemic Adult	Anemic Senior	Reference Interval
BUN (mg/dL)	31.17 ± 5.23	33.10 ± 9.88	53.77 ± 60.25	59.50 ± 43.36	19–34
Creatinine (mg/dL)	1.48 ± 0.10	1.60 ± 0.32	2.60 ± 2.66	3.90 ± 5.13	0.9–2.2
ALT (U/L)	72 ± 29.09	69.57 ± 42.47	88.50 ± 88.80	89.00 ± 80.58	25–97
ALP (U/L)	48.22 ± 16.38	45.88 ± 19.07	19.30 ± 10.67	42.56 ± 38.77	0–45
Total protein (g/dL)	8.96 ± 0.81	8.39 ± 0.59	8.94 ± 2.75	8.88 ± 1.55	6–7.9
Albumin (g/dL)	3.94 ± 0.16	3.78 ± 0.29	3.24 ± 0.44	2.99 ± 0.39	2.8–3.9
Cholesterol (mg/dL)	110.44 ± 20.82	143.13 ± 47.97	144.20 ± 45.35	156.11 ± 57.61	71–156
Triglyceride (mg/dL)	116.56 ± 49.59	148.25 ± 115.69	120.40 ± 93.92	100.89 ± 50.23	27–94

Data are expressed as mean ± standard deviation. Abbreviations: BUN = blood urea nitrogen, ALT = alanine transaminase, and ALP = alkaline phosphatase.

## Data Availability

The data presented in this study are available in the article. Further inquiries can be directed to the corresponding author.

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
