# Peer review of "Feline Erythrocytic Osmotic Fragility in Normal and Anemic Cats—A Preliminary Study"

_vetsci, 2025, doi:10.3390/vetsci12030236_

Round 1
Reviewer 1 Report
Comments and Suggestions for Authors
Please look at the attached file.

Please look at the attached file.
Reviewer 2 Report
Comments and Suggestions for Authors
The present study investigated the red blood cell membrane strength and characteristics in healthy and anemic cats. The results have shown that the cell size and density vary with age and healthy status, whereas the erythrocyte osmotic fragility remains consistent between the groups. Following are some comments on the manuscript.
English: The language of this manuscript should be reviewed by a native English speaker.
Introduction: The logic of introduction is a mess. What is the basic problem that your research focuses on and is done to solve? What is the superiority of your research compared to other researches?
L51-53: What is the meaning of those tests are included? For testing the osmotic fragility? The test which you used in this study should be introduced with more details.
L63: The authors have indicated that there are some reports about the increase of erythrocytic osmotic fragility in anemic cats. This study appears to present results that are already well-documented in existing literature. Then, what is the innovation and significance of this study?
L66-70: This long sentence is irrelevant to your study. This study focused on healthy and anemic cats. Therefore, more introduction is required for anemia.
L95: More details regarding laboratory testing are required. e.g. the details of kits and equipment for the test.
Figures: The sample number of each column should be provided.
L214: The authors should discuss more about the relationship between kidney disease and osmotic fragility.
L224-225: The authors have indicated that some anemic cats exhibit very high osmotic fragility, whereas some exhibit low values. Could the authors present the data? e.g. show the exact value of osmotic fragility and the cause of anemia of all individuals.
In addition, the authors should discuss more about this result. e.g. find more references about the link between osmotic fragility and anemia with different causes, not only in cats, but also in humans and other mammals.
Discussion: The authors should add a paragraph about the implications of this study on veterinary in the end of discussion.
Comments on the Quality of English LanguageThe language of this manuscript should be reviewed by a native English speaker.
Reviewer 3 Report
Comments and Suggestions for Authors
Dear authors. After carefully evaluating your work “Study of Feline erythrocytic osmotic fragility in normal and anemia cats”, please find below some comments.-
English language.- This manuscript needs a really in-depth re-writing and re-organization. Hope you consider to consult with a native speaker, since in the present form is nearly incomprehensible.
Introduction.-I would really recommend re-write the entire introduction. I have not understood the rationale exposed here. The English is kind of strange and the flow of exposition is really lacking.
Line 82.- “hematologic parameters were within normal limits 20”. Do not understand this “20”.
Please clearly describe in detail the physical examination performed (authors are depending on this to classify their healthy patients as such).
Line 84.- At least one. This is kind of problematic. This means that a cat with weight loss is automatically classified as anemic (which is not entirely correct). The same for a cat with only depression or only tachycardia.
Line 87.- Concerning the detection of animals as anemic (according to lab results).- Was hematocrit used IN ISOLATION to detect anemia? Did the authors evaluate the presence/absence of dehydration (which could have falsely elevated the hematocrit)? Which technique was used to determine the hematocrit (microhematocrit manually? Analyzer?). This is really really important.
Overall, I find really inconsistent and problematic the classification of cats (anemic vs healthy).
Line 93.- “will be shipped” Recommend not to use the future tense here.
Line 96.- “Standard”. In a manuscript about feline hematology, it will be really necessary to elaborate this (analyzer, manufacturer, etc.).
Line 98.- “Plain tube”. Nothing has been said about a plain tube (only about EDTA and heparina ones).
Line 100.- Staining method? Who evaluated these smears? In which works/manuscripts/books was based the morphological evaluation? What were you looking for in this evaluation? Did you perform PCR for Mycoplasma (they are really difficult to detect in smears, they can be latent in bone marrow, etc, etc.).
Line 111.- Avoid the future tense.
Line 119.- Same as previous.
Line 120.- Same as previous.
Line 126.- Plural.
Line 145.- “Majority”. In a scientific manuscript authors should be more precise (how many of them?). Apply also for line 148.
Table 1 is of low value and this data could be summarized in the text (with corresponding in-deep description).
Table 2.- FeLV was the cause of anemia in one cat. Are the authors 100% sure ? (this condition is really frequently concurrent with other ones and rarely the unique cause of anemia). “Blood parasite” (which one? This is not acceptable in a scientific manuscript).
Table 3.- References for RBC are missing. These values are mean but, what about the numbers after ± ??? is this SD??? SEM???).
For every table.- Is there any significant difference between groups????
Line 206.- “an adequate quantity of platelets was observed,” How did you evaluate this?
Line 212.- why and how is correlated mycoplasma with low platelets?
Comments on the Quality of English Language
Would recommend to seek the help of a native English speaker
Round 2
Reviewer 1 Report
Comments and Suggestions for Authors
The manuscript has improved following the changes and I believe it can be accepted in this form.
Author Response
Thank you to the reviewer for taking the time to evaluate our manuscript and for providing valuable recommendations. We truly appreciate your insightful feedback and suggestions, which have helped us inprove the quality of our work.
Reviewer 2 Report
Comments and Suggestions for Authors
The manuscript has been improved.
Author Response

(The authors gave the same response as above.)

Reviewer 3 Report
Comments and Suggestions for Authors
Dear authors,
I have reviewed the new version of your manuscript entitled “Feline Erythrocytic Osmotic Fragility in Normal and Anemic 2 Cats – A Preliminary Study”.
I truly appreciate the effort made in this version of the paper and all the changes incorporated.
Concerning the English language, I certainly can see an improvement in the manuscript. Unfortunately, there are still many shortcomings related to the study design (really low number of animals, differentiation between healthy and non-healthy animals, final diagnosis of these patients -mycoplasmosis, etc.-) which makes me conclude that the scientific soundness is still not adequate. Many of my comments/doubts have been responded to in the letter (but changes have not been incorporated to the manuscript).
Find below some specific comments.-
Line 56.- friability. Is this the term that the authors mean to use?
Line 77.- among many other conditions (such as chronic inflammation, tumors, endocrine diseases, nephropaties, etc.).
Line 89.- RBC density is a really strange term. I cannot think what this means…hematocrit?
Line 97.- Here groups are called 1 and 2; later NA and A…
Line 99.- Sadly, this is still my big concern with this manuscript. I truly do not believe that the initial tests made here are able to correctly classify/differentiate between healthy normal cats and cats with subclinical diseases (many of them with known effects on OF, as the authors mentioned in the introduction). Following the classification here exposed cats with splenomegaly, hyperglobulinemia (not mentioned as determined in biochemistry), lymphocytosis, hyperbilirubinemia (not mentioned as determined in biochemistry) or stress (all of them with OF effects, all of them mentioned in the introduction) but WITHOUT ANEMIA, would be classified as healthy cats. Thus, the “healthy cats” group should be a “non anemic cats” group.
Material and methods.- It is really necessary to clarify that no cat was dehydrated (not enough including this in the response to the reviewer).
Line 128.- Unfortunately, M. felis cannot be discarded using only blood smears. This test has a really low sensitivity and specificity and even specialists routinely working with blood smears in cats can have a bad time diagnosis (even more if the sample is in EDTA or it has been stored during at least 24 hours). Without a PCR, from my point of view (and most clin paths) you cannot discard M. felis (or any other mycoplasmosis) in these patients.
Line 152.- A reference for the meaning of these correlation coefficients?
Line 169.- As previously mentioned, which reference ranges are you using/referencing?. This is necessary.
Line 171.- “some” cats is not adequate in a scientific paper. How many of them?
Line 178.- It would be necessary to explain to the reader (not only to the reviewer in the response) what technique was used to determine the platelet concentration.
Tables 2 and 3 and Figures 1, 2 and 3.- As previously said, what does the value behind the mean stands for????? Standard deviation??? SEM??? For example (from your response in the letter)… 2.2 ± 1.2 years… 2.2 is the mean…but what is 1.2??? standard deviation?? SEM???...What does the black line/bar in each column of the Figures stands for???? SD??? SEM????. This is necessary.
Line 235.- Avoid using “healthy” (as previously mentioned).
Line 247.- As previously mentioned, for me it is problematic to draw any conclusion about mycoplasmosis with this design.
